# An SCM-G2SFCA Model for Studying Spatial Accessibility of Urban Parks

**DOI:** 10.3390/ijerph20010714

**Published:** 2022-12-30

**Authors:** Zexu Zhou, Xuedong Zhang, Mengwei Li, Xuedi Wang

**Affiliations:** 1School of Geomatics and Urban Spatial Information, Beijing University of Civil Engineering and Architecture, Beijing 102616, China; 2Key Laboratory for Urban Spatial Informatics of Ministry of Natural Resources, Beijing 102616, China

**Keywords:** Baidu map navigation data, park Baidu score, supply competition, multiple travel modes, SCM-G2SFCA, spatial accessibility

## Abstract

The urban park is the main leisure and entertainment place in residents’ daily lives. The accessibility of parks is closely related to the physical and mental health of the residents. Although many scholars have conducted a great deal of research on the spatial accessibility of urban parks, they have rarely considered the supply competition among different parks and the impact of multiple travel modes on the spatial accessibility of parks. Therefore, based on Baidu map navigation data, in this paper, the park Baidu score is used to represent the competitive relationship among different parks, and the impact of multiple travel modes on the spatial accessibility of parks is considered. A supply competition and multiple travel modes Gaussian two-step floating catchment area (SCM-G2SFCA) model is established to evaluate the spatial accessibility of the parks in the Wuhou District, Chengdu, China. The results show that (1) compared with traditional methods, the SCM-G2SFCA model can obtain more accurate results using Baidu map navigation data. (2) There are obvious spatial differences in the accessibility distribution of the parks in the Wuhou District, Chengdu, with high accessibility in the south and low accessibility in the north. The Jinyang and Huaxing sub-districts in the southern suburbs have the highest park accessibility and can obtain more adequate park services. The Fangcaojie and Cujin sub-districts in the northern urban areas have the lowest park accessibility and are relatively lacking in park services. The research results of this study have important reference value for the rational planning of urban parks and the improvement of the spatial accessibility of urban parks in the Wuhou District of Chengdu and similar urban areas.

## 1. Introduction

Urban parks are a key element of urban greening and low-carbon emission reduction, and they play an important role in recreation and support urban ecological integrity [1,2]. Numerous studies have shown that parks are the main leisure and entertainment place for residents, are an important medium for residents to interact with the natural environment, and affect the physical and mental health of residents [3,4,5]. The participation of residents in daily exercise in parks can effectively promote physical health, improve physical fitness, reduce the incidence of cardiovascular disease and obesity, and play an important role in the health of residents [6,7,8]. Therefore, studying the spatial distribution of urban parks and the actual service efficiency can help assess whether urban parks meet the actual living needs of residents and can assist relevant departments in planning and improving the distribution of urban parks [9,10]. The spatial distribution of urban parks is usually related to the difficulty of reaching public facilities, and accessibility is generally used to measure the spatial distribution of urban parks. Accessibility assessments measure the access, cost, and ease of access to parks [11]. Therefore, studying the spatial accessibility of urban parks is of great significance to economic development and public health [1,12].

Spatial accessibility refers to the size of the interaction opportunity of each node in the transportation network, which can be measured using the travel distance and time [13]. The level of accessibility determines the closeness of the social transportation and the economic connection between the origin and the destination, and it also reflects the convenience of travel for residents in the area, affecting the economic development of the area and the happiness of residents, which is a key factor for measuring the space fairness of public service facilities [14,15]. Therefore, spatial accessibility has received extensive attention from scholars in China and abroad. From the perspective of passenger flow, Cui et al. used machine learning and linear regression models to evaluate public transportation accessibility and to improve the efficiency of public transportation [16]. Aiming at the road impedance problem, Hou et al. proposed a travel time function based on micro-traffic simulation, using which they more accurately predicted the travel time and evaluated the post-disaster traffic demand in Colorado [17]. To study the relationship between the spatial accessibility and service radius, several scholars have carried out research on distance attenuation and the search radius [18,19,20,21,22,23]. Wang et al. used a distance attenuation function to represent the trend of the weight attenuation with increasing distance, which improved the distribution and supply of medical care [24]. Luo et al. proposed a variable radius based two-step floating catchment area (V2SFCA) method for the single search radius problem, which reduces the accessibility error by changing the search radius [25]. McGrail et al. proposed a dynamic search radius 2SFCA method (D2SFCA), which calculates the facility service capacity based on the population density, to assess the medical accessibility in rural Australia [26]. The abovementioned studies mainly improved the accuracy of the accessibility by simulating the effects of the distance attenuation and search radius. However, several important factors affecting the spatial accessibility have rarely been addressed.

Therefore, to more accurately and comprehensively evaluate the spatial accessibility, several scholars have researched factors that affect spatial accessibility, such as the supply relationships, travel modes, and personal preferences [27,28,29,30,31,32,33]. From the perspective of facility service competition, Wan et al. proposed a three-step floating catchment area (3SFCA) method to evaluate the accessibility of medical spaces in Texas [27]. Luo et al. comprehensively considered the impacts of travel cost and park service capacity on travel choices and proposed a Huff 2SFCA (Huff2SFCA) method, which effectively alleviated people’s demand for park services [28]. Shao et al. proposed a supply and demand adjustment 2SFCA (SDA-2SFCA) method according to the number of doctors and the demand population, which better reflects the actual supply and demand for medical services [29]. Considering the impact of travel behavior on park accessibility, Fransen et al. proposed a commuter-based 2SFCA (CB2SFCA) method to evaluate the spatial accessibility in Flanders [30]. Hu et al. assessed the spatial differences between parks and the population in Changchun, China, based on three travel modes (walking, public transportation, and driving) [31]. Considering the travel behavior, mode choice, speed, and travel time, Zhang et al. proposed a travel behavior-based G2SFCA (TB-G2SFCA) method to evaluate park accessibility in Nanjing, China [32]. On this basis, Li et al. evaluated the spatial aggregation pattern of park accessibility in central Nanjing from a community perspective [33]. The abovementioned studies mainly focused on a single factor, such as supply, demand competition or travel modes, to conduct detailed research on accessibility, but few comprehensively consider the impact of multiple factors on the spatial accessibility, and thus, their research results are not sufficiently comprehensive.

In this study, multiple travel modes (driving, public transportation, riding, and walking) were combined to consider the competitive relationship among different parks based on public road data, Baidu map navigation data, and park Baidu score data in the Wuhou District, Chengdu City, Sichuan Province, China. A supply competition and multiple travel modes Gaussian 2SFCA (SCM-G2SFCA) model was established to comprehensively evaluate the accessibility of urban parks in the Wuhou District and to analyze the spatial accessibility of the parks in 17 sub-districts of the Wuhou District. The main contributions of this study are as follows. (1) Baidu map navigation data were used to more realistically simulate people’s travel information. (2) Based on the park Baidu score and various travel modes, an SCM-G2SFCA model was established to evaluate the spatial accessibility of urban parks more realistically and comprehensively. (3) Empirical verification was carried out in Wuhou District, Chengdu City, Sichuan Province, China. The research results provide data support and a reference for the rational planning of the distribution of urban parks in the Wuhou District and similar urban areas and for improving the efficiency of urban park services.

## 2. Materials and Methods

### 2.1. Study Area

The Wuhou District is the urban area of Chengdu City, Sichuan Province, China. It is named after the Wuhou Temple within the territory. The Wuhou District (30°34′–30°39′ N, 103°56′–104°05′ E) is located in the center of Chengdu, faces the Jinjiang District across the river to the east, is adjacent to the Qingyang District to the north, borders the Shuangliu District to the southwest, and is connected to the Gaoxin District to the southeast. It is about 13 km long from east to west and 10 km wide from north to south, with a total area of 76.36 km^2^. The Wuhou District is flat and slightly inclined from northwest to southeast. The administrative area is butterfly-shaped, with an average altitude of 570.50 m (Figure 1). By the end of 2020, the permanent population of the Wuhou District was 1,206,568 people.

### 2.2. Data Sources

The data used in this study mainly included Baidu map navigation data (derived from the Baidu map application programming interface, https://lbsyun.baidu.com/, accessed on 8 August 2022), parks, residential areas, population, road network, and other data (Figure 2). Data for 902 residential areas and 31 parks in the Wuhou District, Chengdu, China, were collected, and the park Baidu score was obtained. The population data were derived from China’s seventh census data, and the permanent population of each residential area was divided according to the areas of the 902 residential areas. Inspired by a previous study [34], the road network, residential area, and park entrances and exits were used to obtain a more realistic spatial accessibility of the urban parks. Taking the centroids of the residential area and park as the origin and destination, the planning path function of the Baidu map navigation data was used to realize travel from the entrances and exits of the residential areas and parks. Therefore, to enable the Baidu map navigation data to select suitable entrances and exits and formulate the best travel route, in this study, the centroids of residential areas and parks were used as the supply locations.

The travel mode is an essential factor in measuring the spatial accessibility of urban parks. Residents’ choices of different travel modes determine the travel time cost from the residential area to the park. According to a previous study [35], we obtained the proportion and frequency of the travel modes in the main urban area of Chengdu, Sichuan Province, China (Table 1). Among them, driving accounted for 25.7% of the total sample, public transportation accounted for 43.2%, riding accounted for 8.6%, and walking accounted for 22.5%. Table 1 shows that public transportation has accounted for the highest proportion and was the primary travel mode.

### 2.3. Methods

In this study, Baidu map navigation data, park data, residential area data, and park Baidu score data were used as the input data. According to the centroid coordinates of the residential areas and parks, the path planning of the Baidu map navigation data was established to determine the travel factors of the four travel modes (driving, public transportation, riding, and walking). Then, according to the population of the residential area, the park area, and the park Baidu score, the supply-to-demand ratio of each park was determined. Based on the G2SFCA method, a supply competition G2SFCA method was established to obtain the spatial accessibility of the parks via the four travel modes. Finally, according to the proportions of the travel modes in the daily travel, a supply competition and multiple travel mode G2SFCA (SCM-G2SFCA) model was established to obtain the spatial accessibility of the parks in the study area. In addition, the spatial accessibility distribution of the urban parks was investigated in depth. Figure 3 presents the technical framework of the spatial accessibility research on urban parks based on the SCM-G2SFCA model.

#### 2.3.1. G2SFCA Method

The G2SFCA method is based on the 2SFCA model and uses the Gaussian function as the distance attenuation function to obtain more realistic accessibility results [21,22].

The 2SFCA method is as follows. (1) For each supply location of a park j, all of the demand locations of residential areas k that are within the threshold travel distance $d_{0}$ from location j are searched, and the supply–demand ratio $R_{j}$ is calculated. (2) For each demand location of residential areas i, all of the supply locations of parks j that are within a threshold travel distance $d_{0}$ are searched, and all of the supply–demand ratios $R_{j}$ are summed to obtain the accessibility $A_{i}$ of the residential area i. According to previous studies [36,37], the formulas of the 2SFCA method are as follows:(1)$$R_{j}=\frac{S_{j}}{\sum_{i\in \left\{d_{ij}\leq d_{0}\right\}}^{k}D_{k}}$$
(2)$$A_{i}=\sum_{j\in \left\{d_{ij}\leq d_{0}\right\}}^{m}R_{j}$$
where i is the demand location of the residential areas, j is the supply location of the parks, $A_{i}$ is the accessibility of demand location i calculated using the 2SFCA method, $d_{ij}$ is the distance between demand location i and supply location j, $R_{j}$ is the ratio of the facility size of supply location j to the population served within the threshold travel distance $d_{0}$, $S_{j}$ is the supply scale of supply location j, and $D_{k}$ is the demand scale of demand location k.

The G2SFCA method uses a Gaussian function as the distance attenuation function within the threshold travel distance of the 2SFCA method. According to a previous study [22], the formulas of the G2SFCA method are as follows:(3)$$R_{j}=\frac{S_{j}}{\sum_{i\in \left\{d_{ij}\leq d_{0}\right\}}^{k}D_{k}\times G\left(d_{ij}\right)}$$
(4)$$A_{i}=\sum_{j\in \left\{d_{ij}\leq d_{0}\right\}}^{m}R_{j}\times G\left(d_{ij}\right)$$
(5)$$G\left(d_{ij}\right)=\{\begin{array}{c} \frac{e^{-\left(1/2\right)\times {\left(d_{ij}/d_{0}\right)}^{2}}-e^{-\left(1/2\right)}}{1-e^{-\left(1/2\right)}},d_{ij}\leq d_{0}\\ \;\;\;\;\;\;\;\;\;\;\;\;0\;\;\;\;\;\;\;\;\;\;\;\;\;\;\;\;\;\;\;\;\;\;,\;d_{ij}>d_{0} \end{array}$$
where $G\left(d_{ij}\right)$ is the Gaussian distance attenuation function from demand location i to supply location j.

#### 2.3.2. G2SFCA Method Based on Supply Competition

Inspired by the study of Shao et al. [29], considering the competitive relationship between different parks, in this study, a supply competition G2SFCA method was developed based on the G2SFCA method. This model replaces the travel distance of the G2SFCA method with the actual travel time from the Baidu map navigation data and uses the park Baidu score to represent the competitive relationship among the different parks. The park Baidu score is a total score evaluated by residents based on the park area, greening, service facilities, and parking location, and it reflects the popularity of each park. In this study, the park Baidu score was taken as the weight of the supply location, which indicates the competitive relationship among the supply locations. The formulas of the supply competition G2SFCA method are as follows:(6)$$R_{j}=\frac{S_{j}\times W_{j}}{\sum_{i\in \left\{t_{ij}\leq t_{0}\right\}}^{k}D_{k}\times G\left(t_{ij}\right)}$$
(7)$$A_{i}=\sum_{j\in \left\{t_{ij}\leq t_{0}\right\}}^{m}R_{j}\times G\left(t_{ij}\right)$$
(8)$$G\left(t_{ij}\right)=\{\begin{array}{c} \frac{e^{-\left(\frac{1}{2}\right)\times {\left(\frac{t_{ij}}{t_{0}}\right)}^{2}}-e^{-\left(\frac{1}{2}\right)}}{1-e^{-\left(\frac{1}{2}\right)}},t_{ij}\leq t_{0}\\ \;\;\;\;\;\;\;\;\;\;\;\;\;\;\;0\;\;\;\;\;\;\;\;\;\;\;\;\;\;\;\;\;\;\;\;\;\;,\;t_{ij}>t_{0} \end{array}$$
where $W_{j}$ is the weight of the supply point, $t_{ij}$ is the travel time from the Baidu map navigation data from demand location i to supply location j, *t*_0_ is the time threshold, and $G\left(t_{ij}\right)$ is the Gaussian attenuation function of the travel time from demand location i to supply location j.

#### 2.3.3. SCM-G2SFCA Model

The residents’ choice of travel mode is also a significant factor affecting the accuracy of the spatial accessibility of urban parks. Inspired by previous studies [32,33], based on the supply competition G2SFCA method, in this study, we comprehensively considered four travel modes, i.e., driving, public transportation, riding, and walking. The proportion of four travel modes is obtained, and the proportion is used as the weight of the accessibility of travel modes. Weighting the accessibility of different travel modes, a supply competition and multiple travel modes G2SFCA (SCM-G2SFCA) model is obtained. The formulas of the SCM-G2SFCA model are as follows:(9)$$R_{j,M_{n}}=\frac{S_{j}\times W_{j}}{\sum_{i\in \left\{t_{ij,M_{n}}\leq t_{0}\right\}}^{k}D_{k}\times G\left(t_{ij,M_{n}}\right)}$$
(10)$$A_{i,M_{n}}=\sum_{j\in \left\{t_{ij,M_{n}}\leq t_{0}\right\}}^{m}R_{j,M_{n}}\times G\left(t_{ij,M_{n}}\right)$$
(11)$$G\left(t_{ij,M_{n}}\right)=\{\begin{array}{c} \frac{e^{-\left(\frac{1}{2}\right)\times {\left(\frac{t_{ij,M_{n}}}{t_{0}}\right)}^{2}}-e^{-\left(\frac{1}{2}\right)}}{1-e^{-\left(\frac{1}{2}\right)}},t_{ij,M_{n}}\leq t_{0}\\ \;\;\;\;\;\;\;\;\;\;\;\;\;\;\;\;\;\;\;\;\;\;\;0\;\;\;\;\;\;\;\;\;\;\;\;\;\;\;\;\;\;\;\;\;\;,\;t_{ij,M_{n}}>t_{0} \end{array},$$
(12)$$A=\sum_{i=1}^{n}W_{M_{n}}\times A_{i,M_{n}}$$
where $t_{ij,M_{n}}$ is the travel time of travel mode $M_{n}$, $W_{M_{n}}$ is the proportion of travel mode $M_{n}$ in daily travel, $G(t_{ij,M_{n}})$ is the Gaussian time attenuation function of travel mode $M_{n}$, $R_{j,M_{n}}$ is the supply-to-demand ratio of travel mode $M_{n}$, $A_{i,M_{n}}$ is the spatial accessibility of travel mode $M_{n}$, and A is the park’s spatial accessibility for the four travel modes.

## 3. Results

### 3.1. Park Spatial Accessibility Analysis of a Single Mode Based on the Supply Competition G2SFCA Method

To explore the differences in the park accessibility among the four travel modes, in this study, the supply competition G2SFCA method was used to compare the spatial accessibility of the parks for each travel mode. According to *the Sichuan Daily* ranking of 100 cities’ travel time in 2021, the average travel time in Chengdu is 39 min, and the average travel distance is 9.1 km. Therefore, in this study, the threshold travel time was set to 39 min. Taking Shenxianshu Park in the center of the study area as the origin, the threshold travel time of the four travel modes in 39 min is shown in Figure 4. As can be seen from Figure 4, at the fastest driving speed, all of the residential areas within the study area can be reached within 39 min. The speed of walking is the slowest, and the residential areas within about 2.7 km around Shenxianshu Park can be reached within 39 min. The speed of riding is moderate, and the residential areas within about 7 km around Shenxianshu Park can be reached within 39 min. Public transportation is greatly affected by the distributions of the subway system and bus lines, and the residential areas along the subway and bus lines can be reached.

Based on the supply competition G2SFCA method, in this study, the spatial accessibility of the park in residential areas was calculated. The kriging interpolation method was used to obtain the spatial distribution of the park accessibility in the study area (Figure 5). Figure 5a shows the spatial distribution of the park accessibility for the driving mode. Residents can get to the park faster because the driving mode has the fastest speed and a shorter travel time. The spatial accessibility of the parks is high, and the accessibility factor is above 1.26. Therefore, all of the residential areas can obtain better park services. Figure 5b shows the spatial distribution of the park accessibility for the public transportation mode. Compared with the driving mode, the public transportation mode is greatly affected by the distributions of the subway system and bus lines, and the traffic conditions in the residential areas around the subway and bus lines are better. The park accessibility around Shiyangchang and Guixi in the southeastern part of the study area is higher, and sufficient park services can be obtained. By contrast, the park accessibility around Fangcaojie and Cujin in the north is the lowest, and park services are relatively scarce. Figure 5c,d show the spatial distributions of the park accessibility for the riding and walking modes. The park accessibility in the northern part of the study area is low, and the park accessibility in the southeast is higher. Residential areas with a park accessibility of 0 cannot obtain park services within 39 min in the northern part of the study area. The results show that there are obvious spatial differences in the distribution of the park accessibility in the study area. For example, the northern part of the study area is the central urban area of Chengdu City. Its traffic patterns are very developed, but the park area is small, the number is scare, and the park accessibility is low. The southern part of the study area contains the two parks with the largest area in the Wuhou District, Jincheng Park, the Guixi Ecological Park, and other parks, and thus, this area has the highest park accessibility and adequate park services.

### 3.2. Analysis of Park Spatial Accessibility Based on the SCM-G2SFCA Model

Residents generally do not use only one travel mode in their daily travel. Therefore, in this study, multiple travel modes were considered to establish an SCM-G2SFCA model to further explore the park spatial accessibility in daily travel and to obtain the spatial distribution of the park accessibility in the study area (Figure 6). Figure 6 shows that the minimum comprehensive park spatial accessibility in the study area is 0.62. These areas are mainly located in the northern and northwestern urban areas, in which park services are relatively scarce. The maximum comprehensive park spatial accessibility is 8.99. These areas are mainly distributed in several large park clusters in the southeastern part of the study area, which have sufficient park services. The comprehensive park spatial accessibility is greater than 0 in the study area. Residents can obtain park services within 39 min of daily travel in the study area.

In order to further explore the advantages of the SCM-G2SFCA model, the Euclidean distance G2SFCA method is used for comparison. In this study, the Euclidean distance G2SFCA method is used to calculate the park accessibility of each residential area, and the Kriging interpolation method is used to obtain the park spatial accessibility in the study area (Figure 7). Compared with the SCM-G2SFCA model (Figure 6), the Euclidean distance G2SFCA method only considers the spatial distance, and cannot consider the impact of the road network on accessibility. For the Huochezhannan sub-district in the middle of the study area, the Euclidean distance G2SFCA method considers that the area is far away from the large parks in the south of the study area, so it is difficult to obtain park services, and the park accessibility is low. In fact, although the Huochezhannan sub-district is far away from the large parks in the south, the area has developed subway and bus lines, and it takes a short time to reach the park, which can provide better park services. Therefore, the Euclidean distance G2SFCA method will lead to some errors, and the SCM-G2SFCA model can solve this problem well. In the SCM-G2SFCA model results, there is high accessibility in the Huochezhannan sub-district, which is in line with the actual situation. Therefore, the SCM-G2SFCA model can obtain more accurate results.

### 3.3. Park Spatial Accessibility Analysis of Sub-Districts in the Wuhou District

The sub-district is the smallest administrative unit in China. Therefore, to quantitatively study the differences in the park spatial accessibility in the different sub-districts, the distribution of the park spatial accessibility was obtained using the weighted average of the park spatial accessibility in the residential areas in the sub-districts (Figure 8). Figure 8a shows the spatial distribution of the park accessibility for the driving mode in the sub-districts. The park spatial accessibility in Jiangxilu, Guixi, Cuqiao, Wangjianglu, Jinyang, and Huaxing is characterized by oversupply, and adequate park services are available. Figure 8b shows the spatial distribution of the park accessibility for the public transportation mode. The park spatial accessibility in Shuangnan, Jitouqiao, Jinyang, and Huaxing in the south is characterized by oversupply, and sufficient park services are available. The park spatial accessibility in Wangjianglu is very good. The park spatial accessibility in eight sub-districts in the northern part of the study area is weak. Furthermore, the Fangcaojie, Cujin, Xiaojiahe, and Hongpailou sub-districts are far from the parks, resulting in a lack of park services in these areas. Figure 8c shows the spatial distribution of the park accessibility for the riding mode. Only the park spatial accessibility in Jinyang and Huaxing in the south is characterized by oversupply, and sufficient park services can be obtained in these areas. However, the park spatial accessibility is low in the northern sub-districts, and park services are relatively scarce. Figure 8d shows the spatial distribution of the park accessibility for the walking mode, which is similar to that of the riding mode, but the spatial distribution of the park accessibility is more uneven. The park services in the southern sub-districts in the study area are sufficient, and the park services in the northern urban areas are seriously lacking. The research results reveal that there are noticeable spatial differences in the park accessibility in the study area, with high accessibility in the south and low accessibility in the north. The northern part of the study area is the central urban area of Chengdu, and the park accessibility is low. By contrast, the southern sub-districts of Jinyang and Huaxing are suburbs of Chengdu and have a high accessibility.

At the sub-district scale, which is the smallest administrative unit, in this study, multiple travel modes were considered to further explore the park accessibility in each sub-district during daily travel. An SCM-G2SFCA model was established to obtain the spatial distribution of the comprehensive park accessibility in the sub-districts (Figure 9). As can be seen from Figure 9, the park accessibility in Jinyang and Huaxing is characterized by oversupply. The park accessibility in Shuangnan, Jitouqiao, Guixi, and Wangjianglu is very good. The park accessibility in Fangcaojie and Cujin is weak. These results indicate that there are noticeable spatial distribution differences in the park accessibility in the Wuhou District, with high accessibility in the south and low accessibility in the north. The park spatial accessibility in Jinyang and Huaxing in the southern suburbs is characterized by oversupply, and more adequate park services can be obtained. However, the park accessibility in Fangcaojie and Cujin in the northern urban area is weak, and the park services are relatively scarce.

Local Moran’I index is used to carry out spatial autocorrelation analysis on accessibility in this study in Wuhou District, Chengdu City, and the cluster type is divided into four types: HH cluster, LL cluster, HL outlier and LH outlier, and not significant (Figure 10). The red area is the HH cluster type, which indicates the residential areas with high accessibility in this area. The blue area is the LL cluster type, which indicates that the area is clustered with residential areas with low accessibility. The grey area is the not significant type, indicating that there are no residential areas with significant high accessibility or with low accessibility. It can be seen from the cluster type in Figure 10 that there are obvious spatial differences in park accessibility. The Shiyangchang and Guixi sub-districts in the south cluster high accessibility residential areas, which can obtain sufficient park services and are the most ideal residential areas. The five sub-districts in the north, Jinyang, Shuangnan, Hongpailou, Jitouqiao and Jiangxilu, lack parks with large area and high service quality, clustering a large number of low-accessibility residential areas, and the park services are scarce. Therefore, relevant departments should plan new parks with high service quality in the Hongpailou sub-district in the north, Cuqiao sub-district in the southwest and Wangjianglu sub-district in the northeast. The number of parks should be increased to meet the park service needs of low accessibility residential areas and better allocate urban park resources.

## 4. Conclusions

Urban parks are the main leisure and entertainment sites in the daily lives of residents, and the spatial accessibility of urban parks is closely related to people’s physical and mental health. Therefore, to comprehensively evaluate the spatial accessibility of urban parks, in this study, Baidu map navigation data and the park Baidu score were used to establish an SCM-G2SFCA model and to evaluate the park accessibility in the Wuhou District, Chengdu City, China. The distribution of park spatial accessibility in the study area was obtained. The main conclusions of this study are as follows.

(1) An SCM-G2SFCA model was developed, and a more realistic park spatial accessibility and distribution in the Wuhou District were obtained. The Baidu map navigation data were used to identify the actual entrances and exits of the parks and residential areas and to achieve access from the entrance and exit. According to the residents’ preferences in choosing parks and travel modes, the spatial accessibility of the urban parks can be evaluated more realistically and accurately.

(2) There are noticeable spatial distribution differences in the park accessibility in the Wuhou District in Chengdu, with high accessibility in the south and low accessibility in the north. Jinyang and Huaxing in the southern suburbs have the highest park accessibility and provide more adequate park services. Fangcaojie and Cujin in the northern urban areas have the lowest park accessibility and lack park services.

Although the method used in this study obtained more accurate spatial accessibility results for the urban parks in the study area, it does not consider the differences in the travel modes of different groups of people, and the elderly may pay more attention to the distance to parks and may tend to preferentially use walking and public transportation to travel in daily lives. At the same time, due to the blocking of districts caused by COVID-19, the local government advocates not to travel across districts, and this study only considers the traveling plans in Wuhou District. When everything returns to normal, the possibility of traveling plans across districts should be considered, and more accurate results will be obtained. In addition, Baidu Map data is used as the only data source in this study, which may have some impact on the results. Therefore, in the future, the mutual cooperation of multi-source map data, the choice of travel modes by different groups and the traveling plans across districts will be considered to more comprehensively assess the accessibility of urban parks and provide data support for urban park planning and construction.

## Figures and Tables

**Figure 1 ijerph-20-00714-f001:**
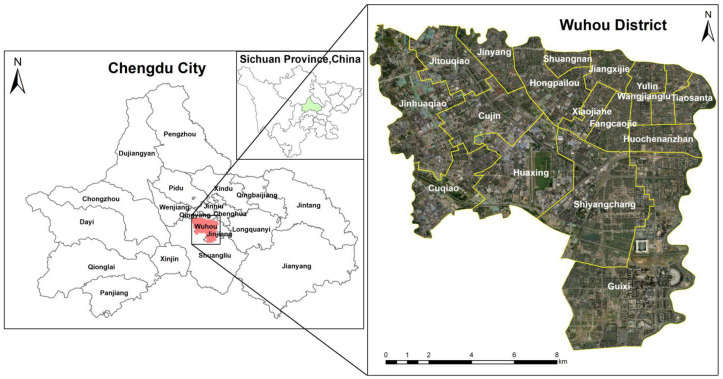
The study Area: the Wuhou District.

**Figure 2 ijerph-20-00714-f002:**
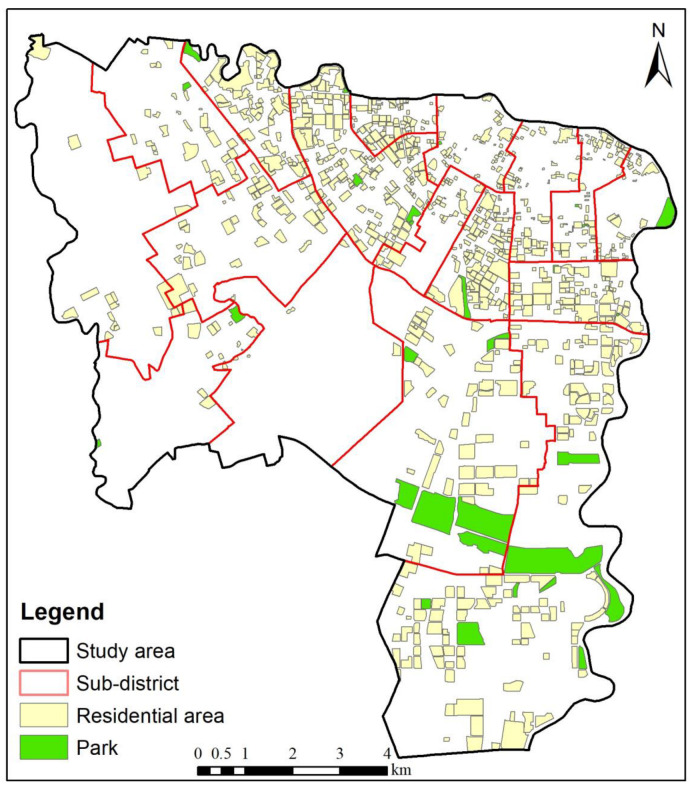
Data sources.

**Figure 3 ijerph-20-00714-f003:**
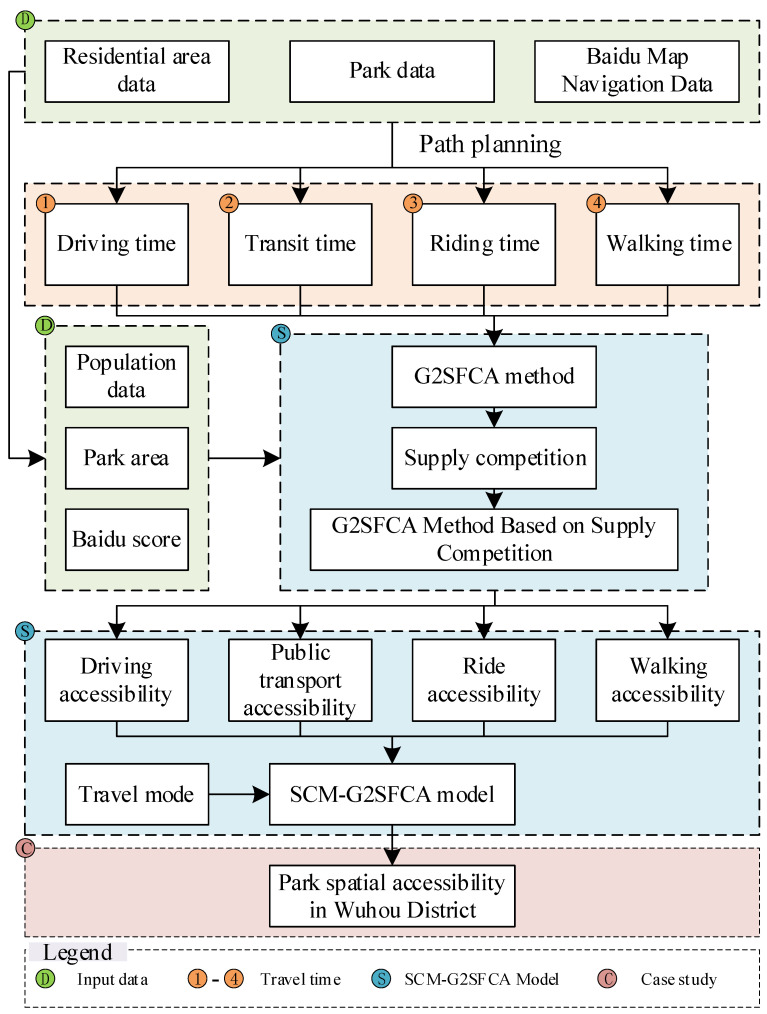
The technical framework of the spatial accessibility research on urban parks based on the SCM-G2SFCA model.

**Figure 4 ijerph-20-00714-f004:**
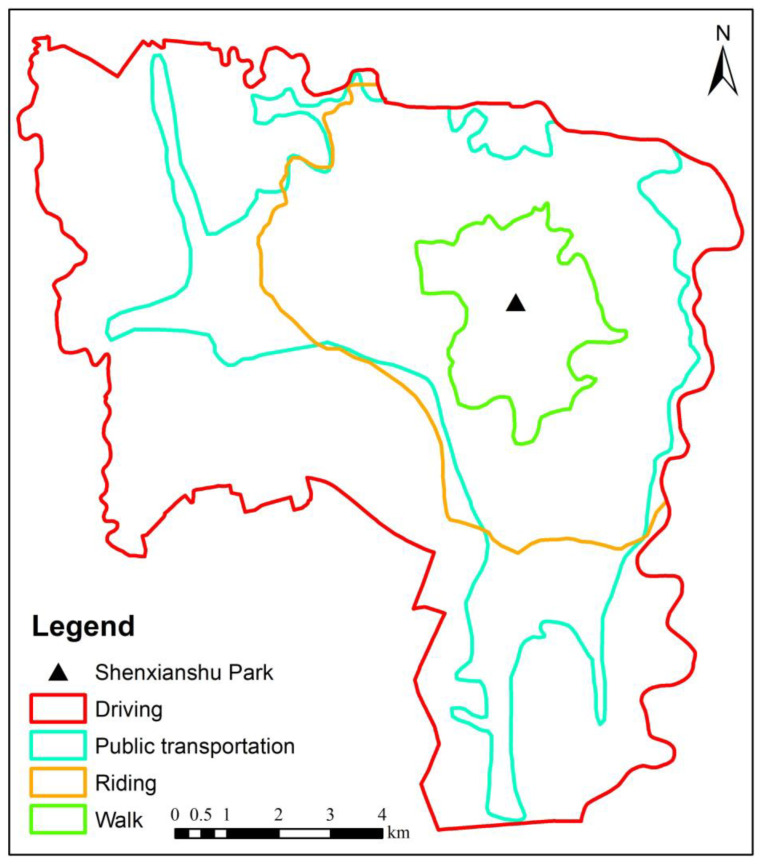
The search radius.

**Figure 5 ijerph-20-00714-f005:**
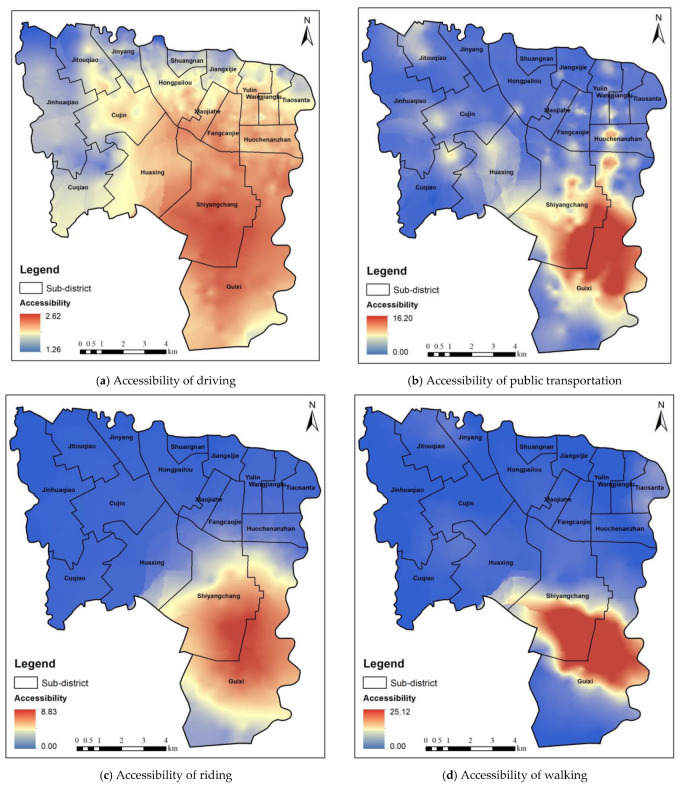
Park accessibility based on supply competition G2SFCA method.

**Figure 6 ijerph-20-00714-f006:**
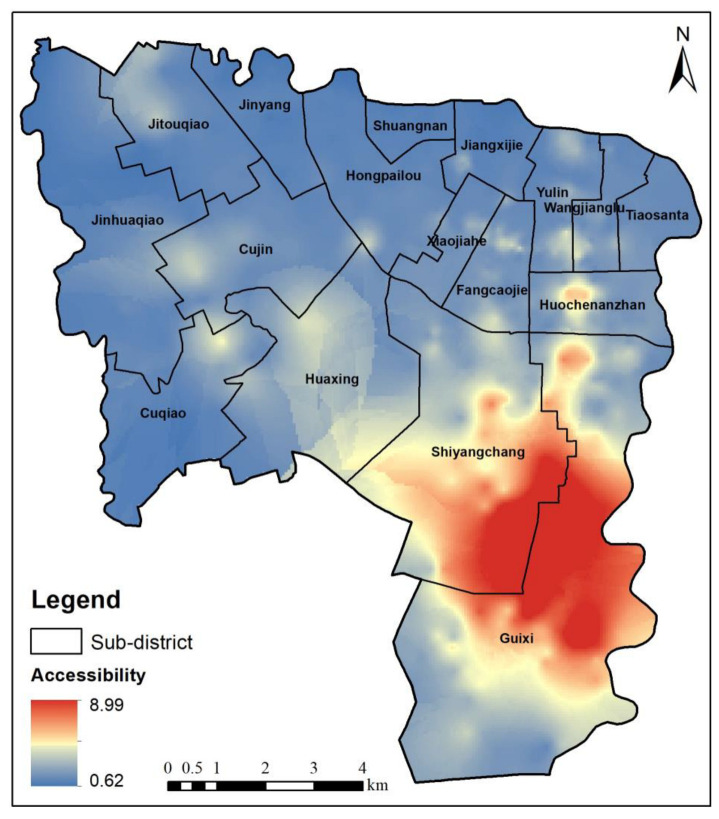
Park spatial accessibility based on SCM-G2SFCA model.

**Figure 7 ijerph-20-00714-f007:**
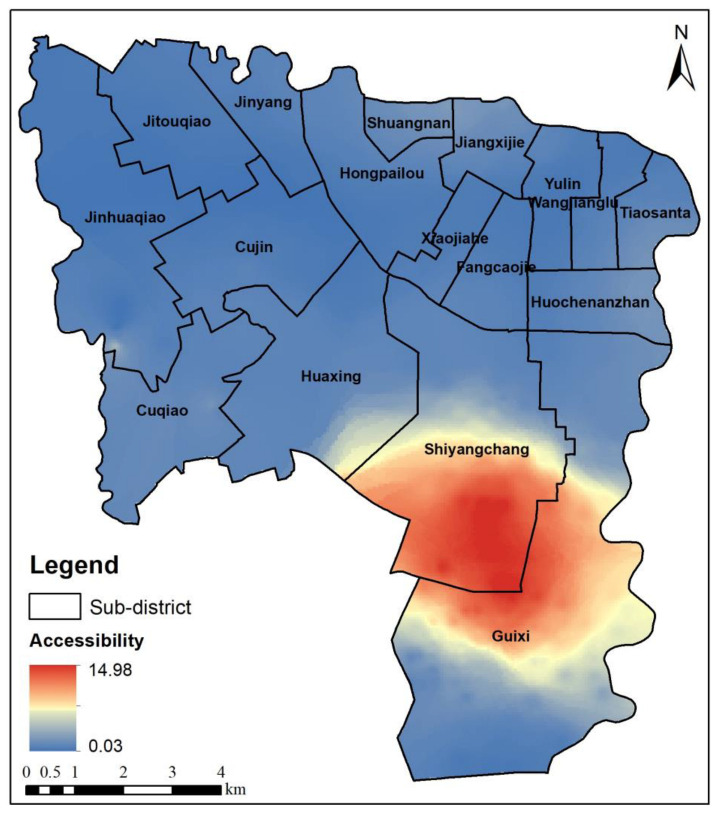
Park spatial accessibility based on Euclidean distance G2SFCA method.

**Figure 8 ijerph-20-00714-f008:**
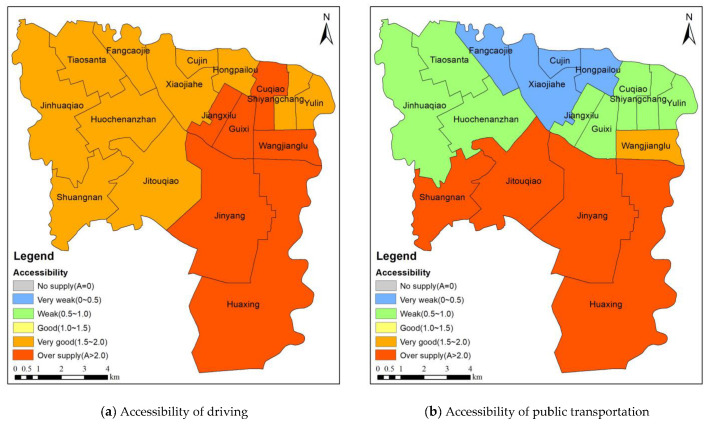
Park spatial accessibility in sub-district based on supply competition G2SFCA method.

**Figure 9 ijerph-20-00714-f009:**
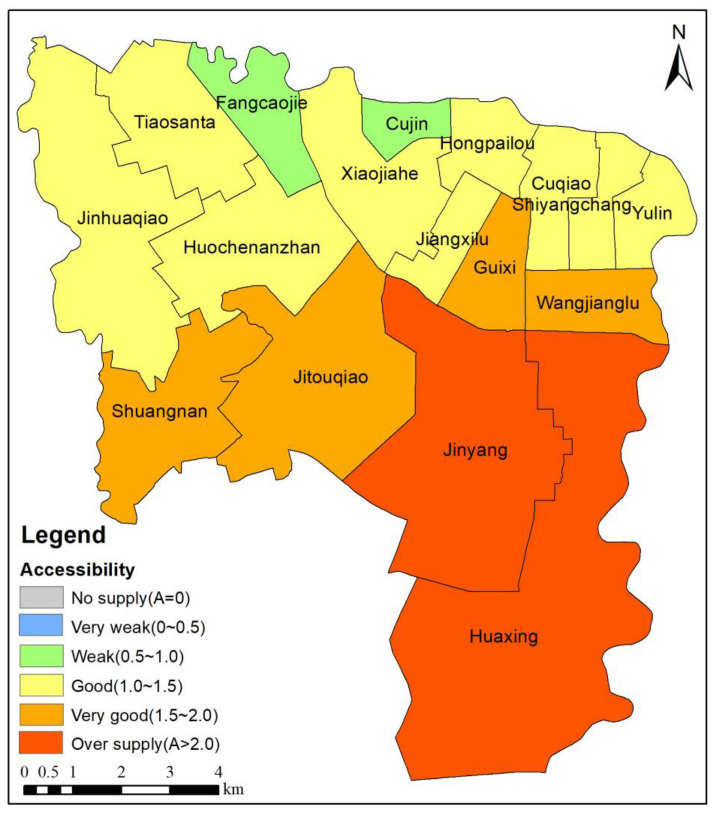
Park spatial accessibility in sub-district based on SCM-G2SFCA model.

**Figure 10 ijerph-20-00714-f010:**
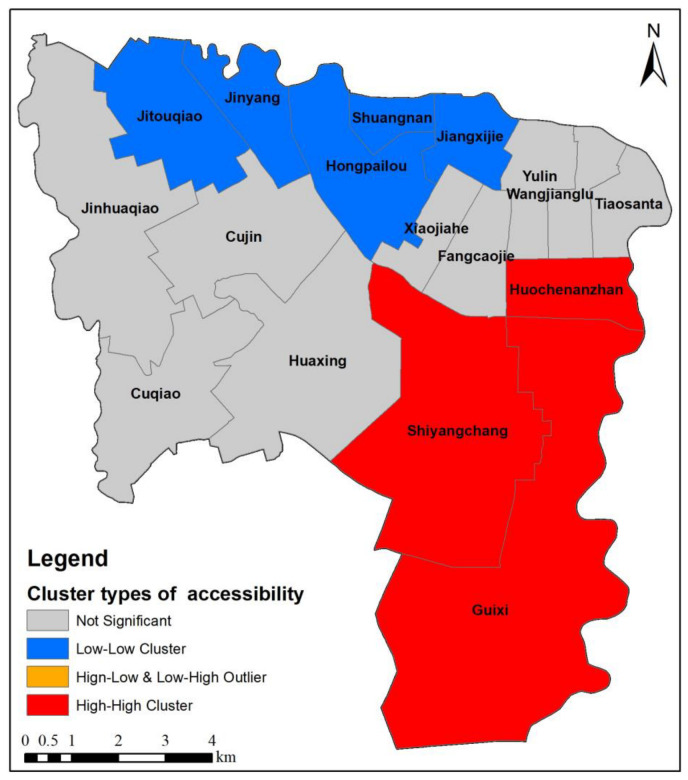
Cluster types of park spatial accessibility.

**Table 1 ijerph-20-00714-t001:** Travel modes.

Travel Mode	Driving	Public Transportation	Riding	Walking
Proportion	25.7%	43.2%	8.6%	22.5%

## Data Availability

The administrative division data used in this article can be found at: https://www.resdc.cn/, accessed on 8 August 2022; Baidu map navigation data can be found at: https://lbsyun.baidu.com/, accessed on 8 August 2022; population data can be found at: http://www.cdwh.gov.cn/, accessed on 8 August 2022.

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
