# Peer review of "An SCM-G2SFCA Model for Studying Spatial Accessibility of Urban Parks"

_ijerph, 2022, doi:10.3390/ijerph20010714_

Round 1
Reviewer 1 Report
The article constructs the SCM-G2SFCA Model to analyse the accessibility of urban parks based on the consideration of supply competition and travel modes, and the results of the study help to promote the rationalisation of urban planning and construction. The study makes use of publicly available data for analysis, the research process is well designed and the findings are somewhat reliable. However, the manuscript still suffers from the following problems that need to be improved.
(1) Only data from Baidu Maps was selected for analysis in the study. Baidu Maps accounts for 51.06% of the mobile phone user segment, while in the field of in-car navigation, Baidu Maps accounts for only about 30%. Is it possible that considering only the data from Baidu Maps may have some impact on the study results? If there may be an impact, it is recommended that the situation be stated or explained in the Limitations section of the study.
(2) Further in-depth analysis of the conclusions drawn from the model calculations is recommended in the Results section, as the current analysis is rather simple.
(3) It is suggested that the Conclusions section should be supplemented with information on how the results of the model calculation and analysis can be used to promote the rationalisation of urban planning and construction, and how to better allocate urban park resources.
Author Response
Please see the attachment:

Reviewer 2 Report
The urban park is the main leisure and entertainment place in residents’ daily lives, and its accessibility is closely related to the physical and mental health of the residents. In this study, Baidu map navigation data and the park Baidu score were used to establish an SCM-G2SFCA model and to evaluate the park accessibility. This work leads to meaningful conclusions. But some issues need to be discussed with the authors.
1. The authors took Wuhou District of Chengdu as the research area. It seems that the northern area is mostly residential areas, and the parks are located in the southern area. However, if you look at it from the perspective of the whole city of Chengdu, the northern residents do not only go to the parks in Wuhou District. Because Chengdu is a whole, they will also go to a park that is closer to another administrative district. From another point of view, the parks in the south of Wuhou are not only visited by residents of Wuhou District, but also by residents of nearby administrative districts. Therefore, the authors must consider this issue, otherwise it would be wrong to draw conclusions.
2. The authors propose a new model, SCM-G2SFCA, but compared with the existing models, what are its advantages? It needs to be discussed and proved.
3. The authors mentioned "we comprehensively considered four travel modes, i.e., driving, public transportation, riding, and walking, and further developed a supply competition and multiple travel modes G2SFCA" in section 2.3.3, but there is no further explanation of how to improve "the supply competition G2SFCA" for adapting multiple travel modes.
4. What is the meaning of WMn in Equation 12? Suggesting that the authors add the definition of WMn in the article
5.In the section 3.1, the threshold travel time is set to 39 minutes, and this time is the time it takes to travel from home to the workplace. In my opinion, people may not spend so much time on the way to the park, so it may be more reasonable that the threshold is smaller than 39.
6. The authors mentioned "The spatial accessibility of the parks is high, and the accessibility factor is above 1.26. Therefore, all of the residential areas can obtain better park services." in section 3.1, why do the residential areas can obtain better park services when the accessibility factor is above 1.26? What's the meaning of the accessibility factor?
7.The maximum value of the accessibility factor shown in figure5(a) is lower than that in figure 5(b). But "Residents can get to the park faster because the driving mode has the fastest speed and a shorter travel time" is mentioned in section 3.1, so the park accessibility for the driving mode should above the park accessibility for the public transportation mode. It is recommended that the authors provide a detailed analysis of this result.
Round 2
Reviewer 2 Report
Thank the authors for considering all my comments. The manuscript is now in a better state.
I strongly suggest that the authors expand the existing research area in future , otherwise the research on only one administrative region will get biased conclusions.